# Sonographic Markers Are Useful for Detection of Early Vascular Deterioration in Children with Overweight/Obesity: Effect of a 1-Year Combined Nutritional Education and Physical Exercise Program

**DOI:** 10.3390/nu15040894

**Published:** 2023-02-10

**Authors:** María Abellaneda Millán, Juan María Morillas-Ruiz, Raúl Ballester Sajardo, Daniel Guillén Martínez, Isabel Morales Moreno, Juan José Hernández Morante

**Affiliations:** 1Department of Nursing, Universidad Católica de Murcia—UCAM, 30107 Murcia, Spain; 2Food Technology and Nutrition Department, Universidad Católica de Murcia, 30107 Murcia, Spain; 3Endocrinology Service, Hospital Virgen de la Arrixaca, 30120 Murcia, Spain; 4Eating Disorders Research Unit, Faculty of Nursing, Universidad Católica de Murcia, 30107 Murcia, Spain

**Keywords:** childhood obesity, nutritional education, physical exercise, carotid intima-media thickness, non-alcoholic fatty liver disease, school nurse

## Abstract

As in adults, childhood obesity is associated with several metabolic disorders; however, there is little information regarding complex obesity-derived complications such as hepatic steatosis or endothelial dysfunction at early ages. Therefore, the objective of the present study was to analyze the effect of a nutritional and physical exercise program on obesity comorbidities in the form of subclinical atherosclerosis and non-alcoholic fatty liver in young children. A total of 148 children completed the study. For nine months (one academic year), children carried out a nutritional and physical activity program of 3 h/week. Anthropometric, clinical, liver ultrasound and carotid intima-media thickness (cIMT) parameters were evaluated. Our results showed that the program significantly improved anthropometrical parameters, especially regarding BMI and body fat (*p* < 0.001 and *p* = 0.004, respectively). The effect on metabolic parameters was limited, although a significant reduction on triglycerides was observed (*p* = 0.009). The education program had a great influence on the echogenic parameters, since the percentage of children with light steatosis decreased from 35% to 4% at the end of the study. In addition, cIMT also decreased significantly in both groups, although there was no statistically significant differences between groups. In conclusion, detection of early complications of obesity through sonographic parameters may allow early intervention, as in the present study, to prevent obesity co-morbidities in children.

## 1. Introduction

Obesity is a complex disease with a multidimensional etiology, pathophysiology and associated comorbidities [1]. Progress of obesity is a multifactorial phenomenon implicating biological, genetic, behavioral, nd environmental factors [2,3]. Above all, the most striking data related to obesity is its elevated prevalence, and even more worrisome, the high prevalence of children with obesity. Attending to recent data, the prevalence of pediatric obesity was 19.7%, which in USA represents about 14.7 million children with obesity [4].

Most complications associated with obesity, such as type 2 diabetes or arteriosclerotic disease, develop gradually over time, and the length of time with obesity increases the risk of developing these disorders [3]. In other words, people with obesity of advanced age have a higher risk of suffering from these alterations. Nevertheless, it has been described that many of these obesity-related comorbidities, such as hypertension, dyslipidemia, fatty liver disease, diabetes, polycystic ovary syndrome (PCOS), sleep apnea, psychosocial problems and others are also present in children [5]. In addition, the early onset of obesity increases the possibility of early morbidity and mortality. In fact, pediatric obesity is an important predictor of adult obesity, and consequently obesity associated disorders [6].

The influence of obesity on hepatic metabolism and its interaction with the cardiovascular system is attracting great attention. Non-alcoholic fatty liver disease (NAFLD) is a clinic-histopathological entity associated with obesity, and it is the most common form of chronic liver disease. Children with obesity and with hepatic steatosis show increased carotid intima-media thickness (cIMT) [7], a marker of arterial thickening in preclinical phases of cardiovascular disease, compared with healthy controls and children with obesity without hepatic steatosis [8]. Consequently, cIMT measurement in NAFLD patients will be an important indicator of carotid atherosclerosis, the major risk factor for cardiovascular diseases [9].

Nowadays, the therapeutic interventions for obesity treatment are mainly based on hypocaloric diets to produce a negative energy balance that, at long-term, may help reduce body weight [10]. However, the latest reviews and meta-analysis regarding the effect of these diets have described moderate to low effectiveness, considering that 90% patients recover their baseline body weight [11]. In addition, the use of hypocaloric diets in children is more complex than that in adults, since an unbalanced diet may interfere with the proper pediatric development [12]. On the other hand, there is limited evidence on the efficacy and security of medications for weight loss in children. Bariatric surgery in young children could also have terrible consequences throughout their lives, so it is generally contraindicated until adulthood [13].

To avoid such consequences, the most promising approach may be to prevent childhood obesity along with an early screening of overweight and obesity [14]. Indeed, the prevention of pediatric obesity through health promotion strategies has been demonstrated to be the most effective public health intervention [15]. These strategies include dietary modification and promotion of regular physical activity. In this regard, family, school and community participation is important for long time results [13]. One potential way of implementing effective school-based interventions may be to involve school nurses, who may be well-suited to deal with childhood obesity, given that they have an ongoing connection with students and families, continual presence in schools and cost-free accessibility to students [16].

Therefore, in the present study, the first aim was to analyze if the early markers of cardiovascular (cIMT) and liver disease, previously described in adolescents, are also present in young children of 4–10 years of age with overweight/obesity. As a secondary objective, a one-academic year health educational intervention was carried out in these young children, to analyze the effectiveness of the educational program, not only on anthropometrical and clinical parameters, but also on obesity comorbidities and their complications in the form of subclinical atherosclerosis (cIMT) and non-alcoholic fatty liver.

## 2. Materials and Methods

### 2.1. Study Design and Participants

For the present study, a quasi-experimental prospective study was designed. The study comprised two groups of children, an intervention group, composed of children with overweight/obesity who underwent a 9-month educational program, based on healthy dietary habits and a physical activity program, and a group of normal weight children who also participated in the same program.

The study was conducted in several schools of Murcia (South-East Spain), with children of 5–10 y. The sample size determination was based on a total of 3272 children in the geographical area, distributed in 15 different schools, regardless of whether they were public or private schools.

In the months prior to the beginning of the academic year (July, August, and September), a recruitment campaign was carried out with advertisements and training sessions, as well as posters and information brochures distributed by different schools in Murcia. Permission was previously requested from the parents or legal guardians to take part in the study. A total of 750 applications were obtained, of which 93 were discarded due to lack of certainty regarding the permission of the legal representative, 402 were rejected for not meeting inclusion criteria and 47 for presenting some health condition that prevented the performance of physical activities. Of the remaining 203 applications, a stratified sampling was carried out to select the final 148 participants.

The selection criteria were children enrolled in the last year of Preschool or the first cycle of Primary School (4–10 years), to take part in the physical exercise program voluntarily by completing an informed consent form by the parents, not performing extra physical activity on a regular basis, as well as not having participated in the previous year in other extracurricular activities offered by the same educational center. On the other hand, those children that followed any kind of specific therapeutical diet and those with any health condition, of any nature, which prevents them from regularly participating in the program, were excluded from the study. Additional exclusion criteria included signs of pubertal development. (Tanner > 2) and remaining under medical treatment that may condition the evolution of the intervention program.

This study respected the Declaration of Helsinki principles. Thus, as a prerequisite to the initial evaluation of the students, the parents and/or legal guardians were informed of the procedures and possible risks of the study through a letter and in previous informative meetings. Before the initial evaluation, parents and/or guardians were required to sign an informed consent form. The study was approved by the Ethics Committee of the Universidad Católica de Murcia (Ethical approval number: #Salud 5–10).

### 2.2. Intervention

After baseline evaluation, children were divided according to their zBMI score. The physical exercise intervention consisted of three hours per week of extra regulated exercise in the afternoon hours. These sessions were conducted on Monday and Wednesday, in 90 min sessions. These children maintained their scheduled physical activity within the physical education lesson in the usual way. The intervention was conducted from the beginning of the academic year (October 2020) to the end of the course (June 2021), which constitutes approximately 9 months of intervention. The clinical evaluation, including anthropometrical, biochemical and other clinical parameters, were evaluated at baseline (September 2020) and at the end of the study (June 2021) by qualified personnel posted to the schools and always prioritizing minimum interference with the development of regular school activities.

The intervention was designed following the recommendations of Janssen et al. [17] and the WHO [18], always supervised by Physical Activity and Sports Sciences professionals in the presence of the school tutors. Exercises were carried out to develop cardiovascular endurance, strength and flexibility, the intensity of which was modified according to the response obtained from the participants. Special emphasis was placed on changing physical activity habits. A total of 68 sessions were held, all of them in three blocks: 15 min of initial warm-up with joint mobility and muscle stretching, 60 min of vigorous intensity that includes jumping, running, receiving and throwing balls, punching bags and exercises for balance, proprioception and agility, and finally 15 min of lower intensity where stretching and lighter intensity activities were carried out again. In addition, a nutritional education program consisted of 14 sessions. These sessions were taught on a fortnightly basis for the first 5 months and monthly for the next 4 months, so that the program had a total duration of 9 months. Each class was conducted via group therapy (groups of 20–25 subjects), with the aim of a greater interaction between the professional and children. Each expositive class was supported by slides, with themes proposed in accordance with the Spanish Society of Nutrition for obesity prevention and treatment [19]. The classes were developed with relevant topics in aspects such as nutrition requirements and recommended foods, daily food delivery and cooking. During the development of the briefings, the participants were encouraged to actively participate, thus increasing the children–professional feedback. The sessions were conducted simultaneously with the physical exercise procedure. However, no participants refused the education program. The sessions were carried out by a multidisciplinary team composed of a registered dietitian with extensive experience in the field of childhood nutrition, two school nurses and a psychologist.

### 2.3. Outcome Variables

The main outcomes of the present study were changes in carotid intima media thickness and hepatic steatosis (measured by liver shear rate and echogenicity). Anthropometrical (zBMI, body fat percentage, waist circumference, etc.), clinical (blood pressure) and other metabolic parameter (lipid metabolism, glucose metabolism and hepatic enzymes) changes were considered as secondary outcomes.

### 2.4. Anthropometrical and Other Clinical Parameters

Body weight was evaluated with children wearing light clothes and barefooted on a TANITA BC-543 scale. Height was obtained with a portable stadiometer (Harpender model, Holtain Ltd., Crymmych, UK). From these data, BMI was estimated. Being a child population, it was necessary to express the BMI values referenced by age and sex according to the WHO reference charts (13), which allows expression of the data in standard deviations. In this way, BMI is expressed as standardized scores or z-score (SD), and classifies children as normal weight (−1 SD to +1 SD), overweight (+1 SD to +2 SD) and obese (≥+2 SD). The waist circumference was measured immediately above the iliac crest, with a flexible tape measure of precision in mm, with the child in standing position and in remote expiration. Body fat percentage was also evaluated by bio-impedanciometry, with the same TANITA BC-543 scale.

Blood samples were obtained after 12 h of fasting by Registered Nurses at 8:00 AM. Samples were immediately kept on ice and translated from school to the reference hospital. All biochemical parameters were analysed with an automated analyzer HITACHI 747–737 (Boehringher Mannheim). To evaluate lipid metabolism, total cholesterol (mg/dL), total triglycerides (mg/dL) and the high-density level and low-density level lipoproteins(mg/dL) were determined. The hepatic enzymes AST(UI/I), ALT(UI/I) and gamma glutamyl transferase(U/L) were also assessed. Fasting plasma glucose(mg/dL) was obtained following the same procedure. Plasma insulin(mg/dL) was determined with an immunoradiometric assay (Medgenix Insulin Kit). HOMA index was estimated according to the following formula: HOMA = fasting plasma glucose (mg/dL) × insulin (mUI/L)/405. A score equal or higher than 3.48 was considered as pathologic.

To estimate blood pressure, a sufficient number of measurements were taken two minutes apart until a difference of less than 5 mmHg was observed between two successive measurements on the right arm, with the child in absolute rest for at least 6 min, comfortable and seated. An automated OMRON^®^ 907 (HEM-907-E) sphygmomanometer with a specific cuff for pediatric patients was used, and the mean of the last three measurements was recorded as the final data. Blood pressure values were also transformed into percentiles to better describe changes from baseline. Those parameters displaying a percentile > 90th were considered as pathologic [20].

### 2.5. Measurement of cIMT and Hepatic Steatosis

The carotid intima media thickness was determined through 4 MHz high frequency ultrasonographic scans from the AcusonS2000 Virtual Touch Tissue Quantification ultrasound equipment (Siemens, Erlangen, Germany). Scans were obtained on the same day of the baseline and final evaluations. Measurements were performed through a grayscale evaluation of the right common carotid artery up to its bifurcation. An evaluation was performed in the longitudinal direction and the thickness of the intima was measured, by color Doppler, taking as reference the anterior edge of the lumen–intima interface and the posterior edge of the media–adventitial interface [21]. The mean value of the data obtained in 3 determinations was calculated.

Point Shear Wave Elastography/Acoustic Radiation Force Impulse (pSWE/ARFI), an ultrasound-integrated elastography technique that studies tissue elasticity by measuring shear rate (SR), was employed. SR is related to the viscoelastic properties of tissues. Measurements were performed with the same equipment for cIMT evaluation (AcusonS2000 Virtual Touch Tissue Quantification), by the same operator. The children were evaluated in the supine position with the right arm in maximum abduction, performing the study between the 6th–7th intercostal spaces, with a standardized technique and without changing the gain. In addition, the presence of fat infiltration was determined and classified as mild (Grade 1), moderate (Grade 2) or severe (Grade 3) following the criteria of Se et al. [21].

### 2.6. Statistical Analysis

All the information obtained from the different tests was collected in a database for later statistical treatment using the SPSS 25.0 software. First, a Kolmogorov-Smirnov normality test was conducted, which confirmed the normal distribution of the variables. A basic descriptive statistic was carried out to evaluate frequencies and data distribution. To evaluate possible statistically significant differences at baseline between children with normal weight and those with overweight/obesity, an independent t-test was conducted. The effectiveness of the physical exercise intervention was performed through a paired t-test, comparing final versus baseline values of outcome variables. To compare the effect of the intervention between groups, an ANOVA analysis was conducted, using group as between-group factor and time as within-group factor. Significance level was established at *p* < 0.050.

## 3. Results

### 3.1. Baseline Characteristics

Table 1 shows the baseline characteristics of both normal BMI children and children with overweight/obesity (zBMI > 1.65). Mean age was similar in both groups. Sex composition was similar and independent of group allocation (*p* = 0.122). In Appendix A, baseline characteristics are described according to the participants’ sex. There were no baseline differences in any characteristics between boys and girls in the control group, although insulin and hepatic shear rate was higher in the girls of the intervention group. As expected by the study design, all obesity-related anthropometrical parameters were significantly higher in the intervention group. Clinical parameters also tended to be worse in the intervention group, as shown by higher levels of hepatic enzymes, plasma insulin and HOMA index. Nevertheless, fasting plasma glucose, blood pressure and echography-related parameters were similar in all children. At baseline, 35% children showed echogenic signals compatible with light steatosis. Hepatic echogenicity was independent of sex (*χ*^2^ = 2.538, *p* = 0.468), but as expected there was a relation between obesity degree and hepatic steatosis (*χ*^2^ = 10.793, *p* = 0.013).

### 3.2. Effect of Educational Intervention on Anthropometrical Parameters

In general, the educational intervention had a more evident effect on the intervention group (Figure 1), since BMI (Figure 1B) and body fat percentage (Figure 1C) were statistically significantly decreased because of the intervention. Concretely, a reduction of 0.35 z-score (CI95%:0.25–0.43) on the BMI was observed in the intervention group, while body fat was reduced by 1.46% (CI95%: 0.43–2.50). On the contrary, waist circumference (Figure 1D) was significantly increased in children with both normal weight and overweight/obesity. Detailed anthropometrical characteristics and intervention effect sizes are shown in Appendix A.

### 3.3. Effect of Health Educational Intervention on Clinical Parameters

There were no statistically significant changes regarding systolic blood pressure in either of the two study groups. However, regarding diastolic blood pressure, although both groups showed lower levels at the end of the intervention, differences were statistically significant only in the control group (*p* = 0.002). Considering percentile values, there was no statistically significant change in the number of participants with systolic pressure < 90th (chi-square = 0.106, *p* = 0.745). However, the effect was more evident regarding diastolic blood pressure, since the number of participants with percentile < 90th increased from 67% to 75% at the end of the intervention (chi square = 6.059, *p* = 0.014).

Regarding lipid metabolism (Figure 2), the most evident effect was noted regarding plasma triglyceride levels, since a statistically significant reduction was observed at the end of the intervention in children with overweight/obesity (Figure 2B). HDL cholesterol fraction (Figure 2C) was also increased in both groups at the end of the intervention, but on this occasion the level of statistical significance was not reached (*p* = 0.145 and *p* = 0.171 in control and intervention groups, respectively).

Carbohydrate metabolism was also evaluated in the present study, but attending to the statistical analysis no significant effect was observed after the intervention. The same situation was observed for the case of liver enzymes (AST, ALT and GGT). (Detailed information about clinical characteristics and effect sizes is shown in Appendix A).

### 3.4. Effect of Health Educational Intervention on Echography-Related Parameters

Figure 3 shows the data regarding cIMT after 9 months of intervention in children with normal weight and overweight/obesity. Despite lower baseline cIMT values in normal weight children, the intervention induced a similar significantly and statistically decrease of cIMT values in both groups (*p* < 0.001 in both cases), and therefore there was no statistically significant effect between groups (inset of Figure 3).

The data derived from liver echogenicity also showed an improvement after the intervention. Concretely, there was an increase in children with normal echogenicity, while there was a similar reduction in children with light echogenicity (Figure 4A). In addition, there were no children with severe echogenicity at the end of the intervention. The shear rate, a measurement of hepatic elasticity, was also reduced because of the intervention in the intervention group, reaching values similar to those of children with normal weight.

## 4. Discussion

The present study was conducted, firstly, to corroborate the presence of early markers of vascular deterioration and hepatic steatosis in young children, which, according to our observations, can be confirmed. In fact, one child showed, at baseline, a liver echogenicity compatible with severe liver damage, reinforcing our initial hypothesis.

Once the presence of these early biomarkers associated with obesity comorbidities had been confirmed, a physical exercise intervention was carried out. At the end of the intervention, BMI and body fat percentage significantly improved. Weight and waist circumference increased, but as a consequence of the own children’s growth. The effectiveness of physical exercise interventions on the improvement of anthropometrical parameters has been intensely described. Previous studies have shown how regular physical exercise within structured lifestyle programs can improve weight status and minimize metabolic risk factors in childhood obesity [22,23,24]. A previous study evaluating the effect of a one-year exercise/lifestyle program showed a significant improvement in obesity markers and glycemic control [24]. Similarly, previous studies have confirmed the benefits of school nutrition/physical activity interventions on body mass index after a [24,25].

Contrary the anthropometrical parameters, the intervention carried out in the present work was not able to significantly modify most of the clinical and metabolic parameters, including blood pressure, biomarkers associated with type 2 diabetes (fasting plasma glucose, plasma insulin or HOMA index) or hepatic enzymes. These observations refute previous results, such as those of Farpour-Lambert et al., since they described that a regular physical activity plan, with aerobic and strengthening exercises lasting 3 months, significantly reduces systemic blood pressure, BMI z-score, and total and abdominal adiposity and increases fat-free mass and cardiorespiratory fitness in pre-pubertal children with obesity. In fact, changes in blood pressure were greater in hypertensive subjects and independent of body fat reduction [22,26]. Nevertheless, although the association between the percentage of body fat and the values of HOMA or insulin has been described [25], there is a great controversy about physical activity behaviors in adolescence, since different reports have described no significant associations between total daily activity and metabolic parameters. The lack of statistically significant effect on metabolic parameters observed in the present work may be related to the baseline characteristics of both groups of children. Although previous studies have confirmed that hypertension is one of the most common complications in children with obesity [27], baseline systolic and diastolic blood pressures were similar and within normal ranges, in spite of the body fat accumulation. This may be associated with the age of the participants, who were only 8 years old, and the studies showing higher blood pressure in individuals with obesity have been conducted in older people.

However, plasma triglyceride levels significantly decreased at the end of the intervention. This improvement may be of great relevance regarding the prospective health of children and may partially explain the improvement observed in sonographic parameters. In fact, as previously described by Kajikawa et al., triglycerides are an independent predictor of endothelial function and lowering their plasma levels may improve endothelial function Similarly, Yusuke et al. have recently shown an association between serum triglyceride levels and development of hypertension, which is in line with the improvement of diastolic blood pressure observed in the present work [28].

In addition to these results, in the present work we wanted to focus on two indicators of comorbidity associated with obesity, for which there is less evidence, such as cIMT and hepatic tissue elasticity. In fact, to our knowledge, no previous intervention analyzing these parameters have been conducted in such young children. In this context, it must be noted that baseline cIMT values were lower in children with normal weight than in children with obesity. Despite this, the intervention had a statistically significant effect, since this parameter was reduced in both groups. A previous study, also conducted on older children with obesity, showed that a physical exercise and nutritional guidance program, for 12 weeks, also reduced BMI, total cholesterol, LDL cholesterol, diastolic blood pressure and carotid intima-media thickness [29]. Children with obesity aged 15 years suffering from increased cIMT, impaired endothelial function and increased cardiovascular risk also showed an improvement of these parameters, and therefore of the cardiovascular risk profile, after a physical exercise intervention [30]. In the study of Woo et al., conducted in children of 9–12 years with obesity, it was shown that diet and physical exercise for 6 weeks improved their vascular dysfunction. Interestingly, despite this short intervention, the changes were sustained at one year [31]. A study carried out in adolescents aged 18 years also showed that reducing arterial stiffness and cIMT from early stages, such as adolescence, can effectively prevent the risks of high blood pressure and overweight and obesity in adulthood [32]. Making changes in the lifestyle habits of young children is a difficult task and in this sense the school environment becomes an ideal place to carry out such changes. A strategy based on increasing aerobic and resistance exercise plans after school, as in in the present work, would be an interesting option. Along these lines, Park et al. showed that an exercise plan for 12 weeks, also performed as afterschool activities, led to a decrease in body composition and a reduction in carotid intima-media thickness [33]. Therefore, considering our data and those of previous authors, it can be assumed that participation in physical activity programs should be encouraged in young children with obesity to prevent premature arterial plaque development and accumulation and, therefore, of atherosclerosis.

This beneficial effect on cIMT was also observed for hepatic-related parameters. In fact, there was an increase in children with normal echogenicity, and a similar reduction in children with mild echogenicity, at the end of the intervention, which reflects that lipid accumulation was reduced as a consequence of the physical activity increase. Most importantly, there were no children with severe echogenicity at the end of the intervention. Interestingly, the shear index was also reduced in both groups after the intervention. Previous studies have described the correlation between liver echogenicity degree detected on ultrasonography and hepatic steatosis [34,35,36], and therefore it may be useful as a marker of the influence of the intervention on liver status. In any case, the present study demonstrates that as anthropometric parameters improved, liver echogenicity and shear index also improved, not only in those children with obesity but also in the group of children with normal weight. This is in line with the observations of El-Koofy et al., which showed that altered anthropometric measurements and intraperitoneal fat were associated with liver echogenicity [37]. In children with obesity with Non-Alcoholic Fatty Liver Disease, arterial function may be restored by improving metabolic risk factors following a 1-year program with diet and physical exercise intervention. It is important to note that, in this study of Pacifico et al., cIMT was not modified at the end of the intervention, in contrast to our observations [29]. It is interesting to note that the imaging parameters were more sensitive than the biochemical markers, where only an improvement in the lipid profile was observed. This could suggest that, in such young children, these imaging techniques may be more useful than biochemical parameters derived from blood draws, which in children can sometimes generate fear or pain.

It is necessary to remark some limitations of the present study. Firstly, there was no additional control for physical activity, so we cannot ensure the extra-physical activity of children. In addition, there was no baseline assessment of diet behaviour and/or physical activity of the participants, which may also influence the effect of the intervention Moreover, we did not follow-up these children for a long-time and the long-term effect of the intervention was masked from our observations; however, considering that previous works with shorter interventions, such as those of Woo et al. [31] and Agbaje et al. [32], have described long-term effectiveness, it is tempting to speculate that our intervention could be also effective long-term. On the other hand, the intervention lasted only 9 months, and therefore it can be considered that longer interventions may have a greater effect. Nevertheless, previous studies have shown that shorter interventions also had similar beneficial effects, which reinforces the suitability of our intervention. Finally, it will be interesting to comment that there is very little information regarding normative values for sonographic parameters and other laboratory markers for such young children. Therefore, although we have observed some improvement on several parameters, we cannot extrapolate our results to clinical practice.

## 5. Conclusions

In conclusion, it can be confirmed that the presence of subclinical atherosclerosis and hepatic steatosis risk factors can already be detected in children with overweight/obesity as young as 8 years old, which highlight the early repercussion of obesity in the cardiovascular system. However, the intervention based on the promotion of physical activity and nutritional education was effective in reducing these risk factors. Considering the difficulty of carrying out lifestyle changes in young children, the school environment becomes very useful, and afterschool interventions such as in the present work become of great interest in order to combat long-term mortality associated with cardiovascular or hepatic diseases. Further studies should continue research on how physical exercise and nutrition education would affect younger children with obesity and what would be the best intervention to improve their health. In the meanwhile, we suggest the evaluation of sonographic parameters for the early detection of obesity-related disorders in children, since these parameters seem to be more sensitive to interventions than other common markers, such as biochemical parameters. In this regard, school nurses emerge as figures with high influence on children’s health, since they can carry out the promotion of health in educational centres, together with the rest of the multidisciplinary team.

## Figures and Tables

**Figure 1 nutrients-15-00894-f001:**
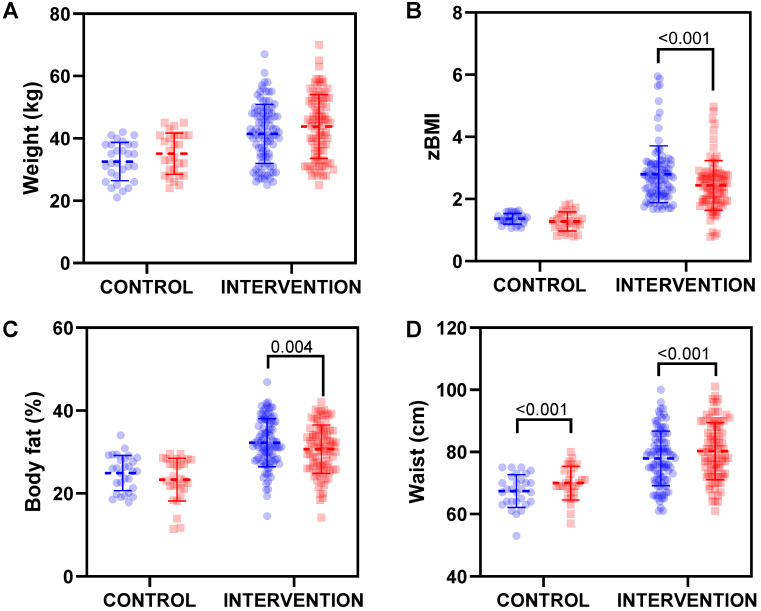
Changes in (**A**) weight, (**B**) BMI z-score (zBMI), (**C**) body fat percentage and (**D**) waist from baseline (blue dots) to the end of the intervention (red dots). The dashed line represents mean value and solid lines represent standard deviation. Statistically significant differences were analysed by a repeated measures ANOVA.

**Figure 2 nutrients-15-00894-f002:**
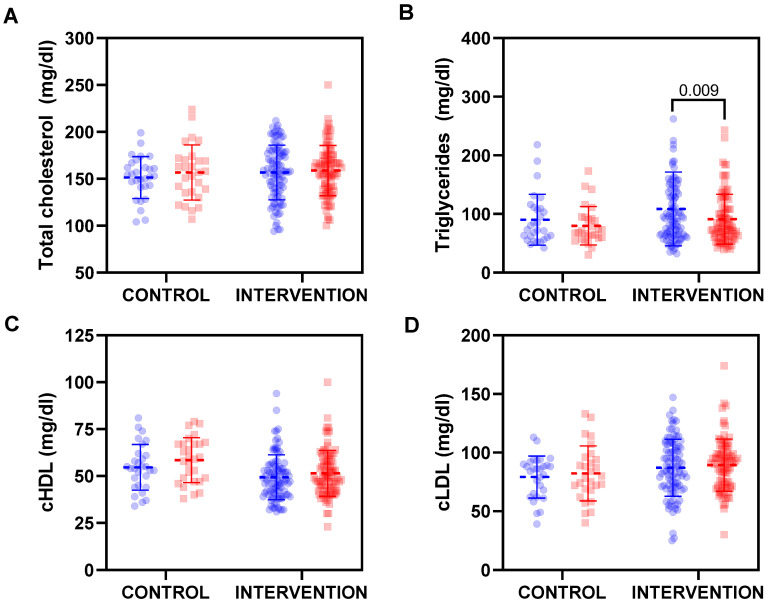
Changes on lipid metabolism parameters: (**A**) total cholesterol, (**B**) triglycerides, (**C**) high-density lipoprotein-cholesterol (cHDL) and (**D**) low-density lipoprotein-cholesterol (cLDL) from baseline (blue dots) to the end of the intervention (red dots). The dashed line represents mean value and solid lines represent standard deviation. Statistically significant differences were analysed by a repeated measures ANOVA.

**Figure 3 nutrients-15-00894-f003:**
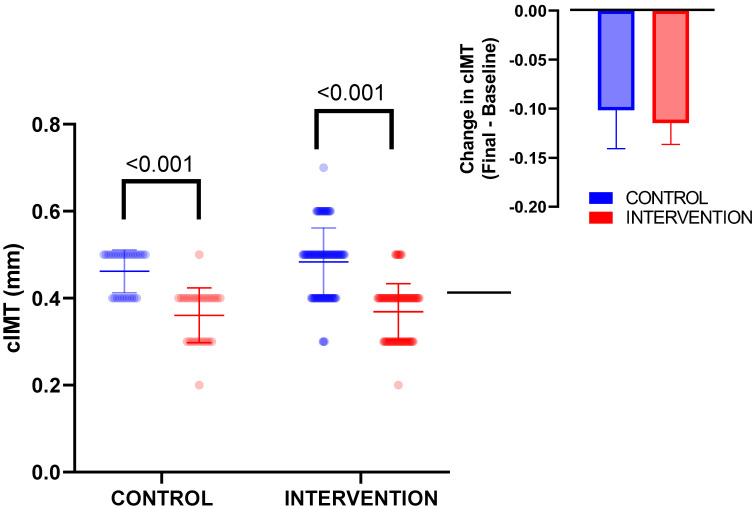
Changes on cIMT in control and Intervention group. The inset represents the cIMT change (final − baseline) in both groups. The intervention significantly decreased cIMT in both groups, although there were no statistically significant differences regarding treatment effectiveness between groups.

**Figure 4 nutrients-15-00894-f004:**
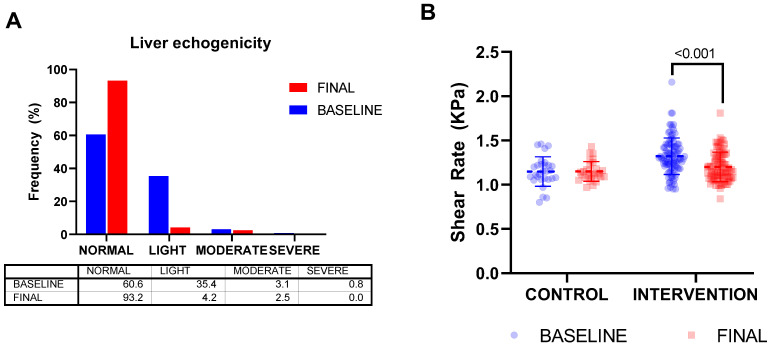
Effect of the intervention on liver echogenicity parameters. (**A**) Frequency distribution of children attending to their liver echogenicity status at baseline and at the end of the intervention. Specific data are detailed in the table below. (**B**) Changes on liver shear rate from baseline (blue dots) to the end of the intervention (red dots). The dashed line represents mean value and solid lines represent standard deviation. Statistically significant differences were analysed by a repeated measures ANOVA.

**Table 1 nutrients-15-00894-t001:** Baseline anthropometrical and clinical characteristics of the participants.

	Control (*n* = 29)	Intervention (*n* = 148)	*p* (*t*)
Age (y)	8 ± 2	8 ± 2	0.881
Weight (kg)	32.6 ± 6.1	41.5 ± 9.5	<0.001
Height (cm)	130.7 ± 9.3	133.2 ± 10.6	0.251
BMI (z-score)	1.36 ± 0.18	2.80 ± 0.91	<0.001
Body fat mass (kg)	8.19 ± 2.31	13.61 ± 4.82	<0.001
Waist circumference (cm)	67.4 ± 5.3	77.9 ± 8.7	<0.001
SBP (mmHg)	106.6 ± 8.1	107.7 ± 10.8	0.601
DBP (mmHG)	69.6 ± 8.6	68.7 ± 8.7	0.615
AST (IU/L)	30.3 ± 5.9	34.1 ± 9.6	0.045
ALT (IU/L)	26.2 ± 6.7	33.0 ± 13.9	<0.001
AST/ALT ratio	1.18 ± 0.22	1.12 ± 0.41	0.410
GGT (IU/L)	14.3 ± 5.8	16.9 ± 5.4	0.028
Total cholesterol (mg/dL)	151.4 ± 22.3	156.8 ± 29.1	0.293
Triglycerides (mg/dL)	90.1 ± 43.3	108.5 ± 63.1	0.145
HDL-chol (mg/dL)	54.6 ± 12.2	49.3 ± 12.0	0.040
LDL-chol (mg/dL)	79.2 ± 17.9	87.1 ± 24.3	0.107
FPG (mg/dL)	84.1 ± 7.1	86.1 ± 5.3	0.115
Insulin (IU/mL)	5.8 ± 2.7	9.9 ± 9.1	<0.001
HOMA index	1.22 ± 0.58	2.15 ± 2.19	<0.001
Carotid IMT (mm)	0.46 ± 0.05	0.48 ± 0.08	0.163
Hepatic Shear Rate (KPa)	1.15 ± 0.16	1.32 ± 0.21	0.445

Data represent mean ± sd. BMI: body mass index. SBP: systolic blood pressure. DBP: diastolic blood pressure. AST: aspartate transaminase. ALT: alanine aminotransferase. GGT: gamma glutamyl transferase. FPG: Fasting plasma glucose. IMT: Intima media thickness.

## Data Availability

The datasets generated and/or analysed during the current study are available in the Mendeley repository [https://data.mendeley.com/datasets/c2d8z7r5z3/1 (accessed on 9 February 2023)].

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
