# Peer review of "Sonographic Markers Are Useful for Detection of Early Vascular Deterioration in Children with Overweight/Obesity: Effect of a 1-Year Combined Nutritional Education and Physical Exercise Program"

_nutrients, 2023, doi:10.3390/nu15040894_

Round 1
Reviewer 1 Report
This is a well-designed and well-written study on a very important topic. The results of the present study provide useful information on the prevention of lifestyle-related diseases in children. I have a small number of suggestions for the authors.
1. Please add a table describing anthropometrical and clinical characteristics of the participants at the end of the intervention, so that the readers can easily grasp the outcome of the study.
2. While the intervention consisted of both diet and exercise in the present study, the relative contribution of each intervention for obesity comorbidities is not clear from the data presented. Thus, the phrase “effect of a 1-year nutritional education program” in the title should be changed to “effect of a 1-year nutritional education and physical exercise program”. In addition, what is the advantage of adding the nutritional educational program for children, whose average age is only 8 years? Please explain this in introduction section.
3. Page 2, line 63. Correct the phrase “more complex that” to read “more complex than that”.
4. Page 2, line 75. Correct the word “whose” to read “who”.
5. Page 4, line 154. I recommend to rephrase the word from “patient” to “participant”.
6. Page 10, line 349-350. The sentence ends mid-sentence.
7. Page 10, line 354-355. I encourage the authors to discuss the significance of the beneficial effect of the intervention on plasma triglyceride levels in the revised manuscript. For example, several studies have reported the relationship between serum triglyceride levels and endothelial function (e.g., Kajikawa et al., Atherosclerosis 2016, 249:70-75) and between serum triglyceride levels and development of hypertension (e.g., Tomita et al., Journal of Hypertension 2021, 39:677-682). Such a discussion would enhance the significance of the results of this study.
Author Response
RESPONSES TO REVIEWER #1
Manuscript: #Nutrients‐2158100 ‐ Sonographic markers are useful for detection of early vascular deterioration in children with overweight/obesity: effect of a 1‐year nutritional education and physical exercise program
REVIEWER #1
This is a well‐designed and well‐written study on a very important topic. The
results of the present study provide useful information on the prevention of
lifestyle‐related diseases in children. I have a small number of suggestions for the authors.
We would like to thank the reviewer’s comments. We have tried to follow his/her
indications as detailed below:
R1#1. Please add a table describing anthropometrical and clinical characteristics of the participants at the end of the intervention, so that the readers can easily grasp the outcome of the study.
We completely agree with the reviewer and have included a new Table S2 as
supplementary information accordingly.
R1#2. While the intervention consisted of both diet and exercise in the present
study, the relative contribution of each intervention for obesity comorbidities is not clear from the data presented. Thus, the phrase “effect of a 1‐year nutritional
education program” in the title should be changed to “effect of a 1‐year nutritional education and physical exercise program”. In addition, what is the advantage of adding the nutritional educational program for children, whose average age is only 8 years? Please explain this in introduction section.
These are some fair comments. On one hand, we have modified according to reviewer #1 and #2 recommendations. On the other hand, it is likely that, in such young children, the effect of the educational intervention was not as effective as in adults, since in children eating habits are strongly influenced by the habits of the family environment, however, as Pérez Rodrigo and Aranceta comment, nutrition education is a key element to promoting lifelong healthy eating and exercise behaviours and should start from the early stages of life [1].
R1#3. Page 2, line 63. Correct the phrase “more complex that” to read “more complex than that”. Page 2, line 75. Correct the word “whose” to read “who”. Page 4, line 154.
I recommend to rephrase the word from “patient” to “participant”. Page 10, line 349‐350. The sentence ends mid‐sentence.
We regret these mistakes. They have been corrected in the revised version of the paper.
R1#4 Page 10, line 354‐355. I encourage the authors to discuss the significance of the beneficial effect of the intervention on plasma triglyceride levels in the revised manuscript. For example, several studies have reported the relationship between serum triglyceride levels and endothelial function (e.g., Kajikawa et al.,
Atherosclerosis 2016, 249:70‐75) and between serum triglyceride levels and
development of hypertension (e.g., Tomita et al., Journal of Hypertension 2021,
39:677‐682). Such a discussion would enhance the significance of the results of this study.
We thank the reviewer’s comment. Effectively, we consider that these references are of great interest, and may improve the discussion section. Therefore, a small
paragraph and these two references have been included in the reviser paper.

Reviewer 2 Report
Dear authors,
thank you for submitting your manuscript to Nutrients. Attached please find my comments:
Title
I find the chosen title a bit confusing:
1. You speak of "echocardiographic" markers. However, no cardiac markers were assessed. Rather use the term sonographic.
2. If I understood correctly, the program lasted 9 months not 12 months
3. The program consisted of nutritional counselling AND physical excercise, hence both of these effects need to be addressed in the title
Introduction
1. Line 52-52 please delete "because of the accumulation of fat"
2. Line 57 please add reference
3. Line 60 please add reference
4. Line 64 "pediatric" instead of "children development" (please change throughout the text)
5. Line 75 "who" instead of "whose"
6. Line 78-80 Please specify the proper aim, I guess you mean "in obese children"?
Methods
1. Please give the trial registration number
2. Please give ethical approval number
3. Was there a baseline assessment of diet behavior and/or physical activity level?
4. In the methods you say "68 sessions of physical activity" were held and "14 sessions of nutritional education program". Therefore I advise to change the title/aim etc. to "effects of a combined physical activity and nutritional educational program"
5. Line 159-162 Please precisely list your main and secondary outcomes
6. Line 165 Use "Body weight" instead of "weight"
7. Line 166 and 175 I guess the word "scale" is missing?
8. Line 180 Please always use units when a new variable is described
9. For lab work and blood pressure: you should additionally use percentiles and describe how abnormal results were defined. As absolute values change during pediatric development.
10. It seems that the measurement of cIMT was not executed in accordance with current guidelines: What protocol was used (please compare with PMID: 25555270)? Was cIMT tracked at end-diastole? Did the authors use ECG to differ between end systole and end diastole? How many loops were recorded? Was the investigator blinded? How did you measure cIMT through color Doppler? The cited reference 18 does not fit. Please add a figure how cIMT was measured.
Results
1. Was there a dropout rate? How many "classes" were visited by study participants?
2. Please add data on sex in Table 1.
3. Please add percentiles for lab work and blood pressure. How many subjects displayed abnormal results (dyslipidemia, arterial hypertension)?
4. Figure 1/2 percentiles should be used for all parameters
5. Was the prevalence of children with abnormal results (dyslipidemia, arterial hypertension) lower after the program?
Discussion
1. I suggest using subheadings
2. Elaborate on the newly added results
3. Elaborate on limitations
Author Response
RESPONSES TO REVIEWER #2
Manuscript: #Nutrients-2158100 - Sonographic markers are useful for detection of early vascular deterioration in children with overweight/obesity: effect of a 1-year nutritional education and physical exercise program
REVIEWER #2
Comments and Suggestions for Authors:
Dear authors, thank you for submitting your manuscript to Nutrients. Attached please find my comments:
Title
R2#1. I find the chosen title a bit confusing: You speak of "echocardiographic" markers. However, no cardiac markers were assessed. Rather use the term sonographic. If I understood correctly, the program lasted 9 months not 12 months. The program consisted of nutritional counselling AND physical excercise, hence both of these effects need to be addressed in the title.
As the reviewer comment, the original title may lead to confusion. We have changed it according to the reviewer’s suggestions.
R2#2. Introduction.
Line 52-52 please delete "because of the accumulation of fat". Line 64 "pediatric" instead of "children development" (please change throughout the text). Line 75 "who" instead of "whose"
We sincerely regret these mistakes. They have been modified in the revised version of the paper
R2#3. Line 57 please add reference. Line 60 please add reference
We apologize for the lack of information. We have included the following references (please note that the line numbers may have changed.):
Reference for Line 57: Wang C, Cai Z, Deng X, Li H, Zhao Z, Guo C, Zhang P, Li L, Gu T, Yang L, Zhao L, Wang D, Yuan G. Association of Hepatic Steatosis Index and Fatty Liver Index with Carotid Atherosclerosis in Type 2 Diabetes. Int J Med Sci. 2021 Jul 23;18(14):3280-3289. doi: 10.7150/ijms.62010.
Reference for Line 60: Yumuk V, Tsigos C, Fried M, Schindler K, Busetto L, Micic D, Toplak H; Obesity Management Task Force of the European Association for the Study of Obesity. European Guidelines for Obesity Management in Adults. Obes Facts. 2015;8(6):402-24. doi: 10.1159/000442721. Epub 2015 Dec 5. Erratum in: Obes Facts. 2016;9(1):64.
R2#4. Line 78-80 Please specify the proper aim, I guess you mean "in obese children"?
Effectively, as the reviewer comment, it will be more appropriate to describe in the aim that the study was focused in children with obesity. Therefore, it has been modified in the revised paper.
R2#5. Methods
R2#5.1. Please give the trial registration number
We are somewhat confused about this reviewer’s comment. In our opinion, the present study was focused on the detection of early markers of vascular deterioration on young children, therefore, we believe that the present work is not applicable to be registered as a trial. Although we have previously registered some trials (for instance, see: https://clinicaltrials.gov/ct2/show/NCT03755674), we consider that the present work is a prospective longitudinal study, and will not meet the criteria to be registered (for further information, please refer to: https://clinicaltrials.gov/ct2/manage-recs/fdaaa#WhichTrialsMustBeRegistered)
R2#5.2. Please give ethical approval number
We regret the lack of information in the original version paper. The Ethical Approval Number has been included in the Methods section of the revised paper, and the original document will be delivered under reasonable request.
R2#5.3. Was there a baseline assessment of diet behavior and/or physical activity level?
It is a good remark by part of the reviewer. Unfortunately, there was no such baseline evaluation. As previously commented, the main objective was the detection of these parameters, and although a biochemical characterization was carried out, there was no previous information. As this is an interesting remark, we have included this information in the limitation section of the revised paper.
R2#5.4. In the methods you say "68 sessions of physical activity" were held and "14 sessions of nutritional education program". Therefore I advise to change the title/aim etc. to "effects of a combined physical activity and nutritional educational program"
We agree with the reviewer, and the title has been adapted accordingly.
R2#5.5. Line 159-162 Please precisely list your main and secondary outcomes
Again, we regret the lack of information. It has been included in section 2.3 of the revised paper.
R2#5.6. Line 165 Use "Body weight" instead of "weight". 7. Line 166 and 175 I guess the word "scale" is missing? Line 180 Please always use units when a new variable is described
We thank the reviewer’s comments. We have modified it.
R2#5.7. For lab work and blood pressure: you should additionally use percentiles and describe how abnormal results were defined. As absolute values change during pediatric development.
It is a fair comment by the reviewer. We have explained this issue in the methods section and included the information in the results section.
R2#5.8. It seems that the measurement of cIMT was not executed in accordance with current guidelines: What protocol was used (please compare with PMID: 25555270)? Was cIMT tracked at end-diastole? Did the authors use ECG to differ between end systole and end diastole? How many loops were recorded? Was the investigator blinded? How did you measure cIMT through color Doppler? The cited reference 18 does not fit. Please add a figure how cIMT was measured.
The sonographic parameters were evaluated by professional radiologist with many years of clinical experience. They followed the protocol described in the manufacturer’s manual, which are those described in [1]. We can confirm that all radiologists were blinded for the study. Radiologists did not know participants allocation and any other personal information.
- James H Stein, Claudia E Korcarz, R Todd Hurst, Eva Lonn, Christopher B Kendall, Emile R Mohler, Samer S Najjar, Christopher M Rembold, Wendy S Post; American Society of Echocardiography Carotid Intima-Media Thickness Task Force. Use of carotid ultrasound to identify subclinical vascular disease and evaluate cardiovascular disease risk: a consensus statement from the American Society of Echocardiography Carotid Intima-Media Thickness Task Force. Endorsed by the Society for Vascular Medicine. J Am Soc Echocardiogr. 2008 Feb;21(2):93-111; quiz 189-90. doi: 10.1016/j.echo.2007.11.011.
R2#6.1 Results. Was there a dropout rate? How many "classes" were visited by study participants?
We did not include this information because only one child in the intervention group dropped out the study. In general, students attended all classes, and with rare exceptions due to illness or other causes, students participated in all sessions.
R2#6.2. Please add data on sex in Table 1.
This information has been included as Table S1, and a brief paragraph has been included in the results section.
R2#6.3. Please add percentiles for lab work and blood pressure. How many subjects displayed abnormal results (dyslipidemia, arterial hypertension)?
The comment of the reviewer is quite interesting. We have included percentile values in the results section, obtaining a similar conclusion. Namely, there was no change regarding systolic blood pressure, but the number of participants that improved diastolic blood pressure was significant. However, the percentile values for laboratory markers are not clearly stated for a so young population, and in our opinion, information and including this information could add little value to the data.
R2#6.4 Was the prevalence of children with abnormal results (dyslipidemia, arterial hypertension) lower after the program?
As commented above, this information has been included regarding blood pressure level.

Round 2
Reviewer 2 Report
/
Author Response
Thank you for your time.